# Fetal Ultrasound and Magnetic Resonance Imaging Abnormalities in Congenital Cytomegalovirus Infection Associated with and without Fetal Growth Restriction

**DOI:** 10.3390/diagnostics13020306

**Published:** 2023-01-13

**Authors:** Kenji Tanimura, Akiko Uchida, Mizuki Uenaka, Hitomi Imafuku, Shinya Tairaku, Hiromi Hashimura, Yoshiko Ueno, Takumi Kido, Kazumichi Fujioka

**Affiliations:** 1Department of Obstetrics and Gynecology, Kobe University Graduate School of Medicine, Kobe 650-0017, Japan; 2Department of Obstetrics, Hyogo Prefectural Kobe Children’s Hospital, Kobe 650-0047, Japan; 3Department of Radiology, Kobe University Graduate School of Medicine, Kobe 650-0017, Japan; 4Department of Pediatrics, Kobe University Graduate School of Medicine, Kobe 650-0017, Japan

**Keywords:** congenital cytomegalovirus infection, diagnostic accuracy, fetal growth restriction, ultrasound, prenatal diagnosis

## Abstract

Congenital cytomegalovirus infection (cCMV) can cause fetal growth restriction (FGR) and severe sequelae in affected infants. Clinicians generally suspect cCMV based on multiple ultrasound (US) findings associated with cCMV. However, no studies have assessed the diagnostic accuracy of fetal US for cCMV-associated abnormalities in FGR. Eight FGR and 10 non-FGR fetuses prenatally diagnosed with cCMV were examined by undergoing periodic detailed US examinations, as well as postnatal physical and imaging examinations. The diagnostic accuracy of prenatal US for cCMV-associated abnormalities was compared between FGR and non-FGR fetuses with cCMV. The diagnostic sensitivity rates of fetal US for cCMV-related abnormalities in FGR vs. non-FGR fetuses were as follows: ventriculomegaly, 66.7% vs. 88.9%; intracranial calcification, 20.0% vs. 20.0%; cysts and pseudocysts in the brain, 0% vs. 0%; ascites, 100.0% vs. 100.0%; hepatomegaly, 40.0% vs. 100.0%; splenomegaly, 0% vs. 0%. The diagnostic sensitivity of fetal US for hepatomegaly and ventriculomegaly in FGR fetuses with cCMV was lower than that in non-FGR fetuses with cCMV. The prevalence of severe long-term sequelae (e.g., bilateral hearing impairment, epilepsy, cerebral palsy, and severe developmental delay) in the CMV-infected fetuses with FGR was higher, albeit non-significantly. Clinicians should keep in mind the possibility of overlooking the symptoms of cCMV in assessing fetuses with FGR.

## 1. Introduction

Cytomegalovirus (CMV) is the most common cause of congenital infection in humans. The prevalence range of congenital CMV infection (cCMV) in newborns is reportedly 0.2–2.4% [1]. Ten to fifteen percent of infected fetuses show symptomatic cCMV at birth, and 90% of the surviving infants develop severe long-term neurological sequelae [2]. Recently, early intervention with oral valganciclovir (VGCV) was reported to improve neurological outcomes in children with symptomatic cCMV [3,4]. Therefore, the prenatal identification of newborns at a high risk for cCMV is clinically important.

Previously, we found that fetal abnormalities associated with cCMV based on ultrasound (US) testing were strong predictive factors for cCMV, regardless of whether the mothers were suspected of primary CMV infection [5,6]. US findings suggestive of cCMV include fetal growth restriction (FGR), ventriculomegaly, intracranial calcification, cysts or pseudocysts in the brain, microcephaly, ascites, hepatosplenomegaly, hyperechoic bowel, and placentomegaly. FGR is among the most frequently observed US findings in fetuses with cCMV [5,7,8,9], although it is not specific to cCMV. Previous studies have demonstrated that the routine maternal toxoplasma, rubella, CMV, and herpes simplex virus (TORCH) test for fetuses with isolated FGR is not justified [10,11]; however, maternal CMV antibody testing should be considered for fetuses with multiple abnormalities associated with cCMV based on US testing [12,13]. Conversely, we have encountered a fetus who was thought to have FGR alone on prenatal US examinations but was later found to have cCMV and multiple morphological abnormalities related to cCMV after birth. The accurate identification of morphological abnormalities associated with cCMV in fetuses with FGR by clinicians and sonographers has not been established. To the best of our knowledge, no study has compared the diagnostic accuracy of prenatal US examinations in detecting abnormalities from cCMV between fetuses with FGR and those without FGR. In addition, previous studies have demonstrated that the diagnostic sensitivity of US screening for cCMV was much lower than that of targeted US examinations performed with the knowledge of fetal CMV infection [14,15,16]. Since October 2009, we have been conducting a single-center trial on fetal therapy administered by immunoglobulin (Ig) injection into the peritoneal cavity of an affected infant or into maternal blood for fetuses with symptomatic cCMV, which was diagnosed by the presence of both imaging findings associated with cCMV and CMV-DNA in the amniotic fluid obtained by prenatal amniocentesis [17]. Therefore, we often managed fetuses with prenatally confirmed cCMV with a positive CMV-DNA PCR in amniotic fluid. In this study, we described the associated fetal ultrasound/MRI abnormalities in CMV-infected fetuses with and without FGR. In addition, we compared the diagnostic performance of targeted US examinations according to the FGR status to determine if there are differences in US findings associated with cCMV in fetuses with growth restriction.

## 2. Materials and Methods

This retrospective cohort study was approved by the Institutional Review Board of Kobe University Hospital (Reference Number 928). Fetuses delivered at the university hospital who were prenatally diagnosed with symptomatic cCMV through imaging findings associated with cCMV and positive amniotic fluid CMV-DNA PCR were included. Fetuses in pregnancies that ended in induced abortion or those delivered at other hospitals were excluded. Eighteen fetuses who were prenatally diagnosed with symptomatic cCMV between October 2009 and October 2022 were enrolled. All fetuses underwent detailed US examinations for the detection or follow-up of imaging findings associated with cCMV at least once a week. The following US findings in fetuses and placentas were defined as imaging findings associated with cCMV, as follows: FGR, ventriculomegaly, intracranial calcification, cysts or pseudocysts in the brain, microcephaly, ascites, hepatomegaly, splenomegaly, hyperechoic bowel, and placentomegaly (Figure 1). The US examinations were performed by perinatologists (K.T., M.U., and S.T.) using the Voluson E8 or E10 (GE Healthcare, Milwaukee, WI, USA) or the ARIETTA 60 (Hitachi Aloka Medical, Tokyo, Japan). FGR was defined as an estimated fetal body weight (EFBW) ≤ the 3rd percentile for gestational weeks (GW) [18,19]. Microcephaly was defined as a head circumference (HC) ≤ the mean −3.0 SD for GW [20]. Ventriculomegaly was defined as a lateral ventricular atrial diameter ≥ 10 mm, and classified into three categories: mild (10–12 mm), moderate (12–15 mm), or severe (≥15 mm) [21]. Hepatomegaly and splenomegaly were defined as biometric measurements of the fetal liver and spleen beyond the 95% confidence interval (CI) of the reference values for GW, respectively [22,23]. Placentomegaly was defined as a placenta thickness ≥4 cm in the second trimester or ≥6 cm in the third trimester [24,25].

The mothers of the 18 fetuses underwent magnetic resonance imaging (MRI) examinations using a 1.5T MR system (Gyroscan NT, Philips, Best, The Netherlands). The MRI protocol included axial, sagittal, and coronal T2 weighted images (T2WI) using half-Fourier acquisition single-shot turbo spin-echo, balanced fast-field echo imaging, and axial double-echo gradient-echo chemical shift imaging. Two board-certified radiologists with more than 10 years’ experience (H.H. and Y.U.) retrospectively evaluated the MR images for the following findings: ventriculomegaly, pseudocysts, cortical malformation, cerebellar hypoplasia, and white matter (WM) hyperintensity on T2WI (Figure 2). The two radiologists independently evaluated the MR images, and the final diagnoses were made by consensus. During the interpretation, the radiologists were aware that the fetuses had cCMV but were blind to information about fetal US, as well as neonatal clinical findings and outcomes.

All 18 newborns, including 2 cases of early neonatal death, underwent physical and imaging examinations conducted by experienced neonatologists, as well as blood tests, as early as possible after birth. For the 16 live newborns, cCMV was confirmed by a urine CMV-DNA PCR analysis. The definition of ventriculomegaly used for the newborns was the same as that for the fetuses. Microcephaly was defined as an HC ≤ the mean −2.0 SD for newborns of the same age and sex. Intracranial calcification and pseudo cysts in the brain were detected by ultrasounds (Figure 3). Hepatomegaly was defined as, on palpation, a liver edge ≥ 3.5 cm below the right costal margin. Splenomegaly was defined as, on palpation, a tip of the spleen ≥ 2.0 cm below the left costal margin. Furthermore, the 16 live infants underwent physical and auditory brain-stem response examinations every 3 months until 1 year of adjusted age and every 3–12 months thereafter. The neurological outcomes at 1.5 and 3 years of adjusted age were assessed using physical examinations and the developmental quotient (DQ) using the Kyoto Scale of Psychological Development [26]. Mild and severe developmental delays were defined as overall DQ scores of 70–79 and <70, respectively.

The diagnostic accuracy rates of prenatal US for morphological abnormalities associated with cCMV were compared between fetuses with and without FGR who had cCMV. In addition, the clinical characteristics of mothers and newborns, fetal US findings, physical findings of newborns, and outcomes of newborns were compared between the FGR and non-FGR fetuses. Differences between the two groups were analyzed using the Mann–Whitney U test and Fisher’s exact test. Statistical significance was considered at *p* < 0.05. All statistical analyses were performed using EZR (Saitama Medical Center, Jichi Medical University, Saitama, Japan), a graphical user interface for R (The R Foundation for Statistical Computing, Vienna, Austria).

## 3. Results

### 3.1. Prenatal Imaging Findings, Postnatal Clinical Findings, and Outcomes for Fetuses with Both cCMV and FGR

Eight fetuses with FGR were prenatally diagnosed with cCMV based on positive CMV-DNA PCR testing of the amniotic fluid. The prenatal US findings, fetal brain MRI findings, and neonatal clinical findings and outcomes are shown in Table 1. Case 3 was referred to our hospital at 20 GW because the maternal TORCH screening was positive for serum CMV IgM. Thus, the case was diagnosed with FGR at 26 GW at our hospital. The remaining 7 pregnant women were referred to our hospital because they had fetuses with FGR together with positive results for serum CMV IgM.

All eight mothers of the fetuses were positive for CMV IgM during pregnancy. Two mothers of the fetuses (cases 1 and 3) were estimated to have primary CMV infections in or before the second trimester, because they had low CMV-IgG avidity index (AI) values (Aisenkai Nichinan Hospital Miyazaki, Japan) (cutoff value: <35% [27]) in the second trimester. The mother of case 2, who had low CMV IgG AI values in the third trimester, could have a primary CMV infection in the second trimester. In the five mothers (cases 4–8), the timing of the CMV infection was unknown.

The median GW at the time of diagnosis of FGR was 24 GW (range, 19–30 GW). The prevalence rates of fetal US findings associated with cCMV in the 8 cases were as follows: ventriculomegaly, 62.5% (5/8); intracranial calcification, 12.5% (1/8); cysts or pseudocysts in the brain, 0% (0/8); microcephaly, 25.0% (2/8); ascites, 25.0% (2/8); hepatomegaly, 25.0% (2/8); splenomegaly, 0% (0/8); hyperechoic bowel, 12.5% (1/8); placentomegaly, 0% (0/8). Two fetuses had FGR only (cases 2 and 3), whereas the remaining 6 fetuses showed other cCMV-associated US findings in addition to FGR. Six of the eight fetuses (cases 1–6) received immunoglobulin (Ig) fetal therapy as reported previously [17,28].

The MRI findings of the fetal brain evaluated retrospectively in this group were as follows: ventriculomegaly, 50.0% (4/8); pseudocysts, 25.0% (2/8); schizencephaly, 12.5% (1/8); cerebellar hypoplasia, 37.5% (3/8); WM hyperintensity, 25.0% (2/8).

The median GW values at delivery and birth weight were 36 GW (range, 31–36 GW) and 1842 g (range, 1255–2192 g), respectively. The prevalence rates of clinical and imaging findings at birth in the eight newborns with both FGR and cCMV were as follows: ventriculomegaly, 75.0% (6/8); intracranial calcification, 62.5% (5/8); cysts or pseudocysts in the brain, 12.5% (1/8); microcephaly, 25.0% (2/8); ascites, 12.5% (1/8); hepatomegaly, 62.5% (6/8); splenomegaly, 0% (0/8); anemia, 25.0% (2/8); thrombocytopenia, 87.5% (7/8). One newborn died at 1 day of age due to having a hypoplastic lung (case 1), while the remaining 7 newborns were alive. The remaining 7 live infants received neonatal therapy with oral VGCV as reported previously [3].

Four of the 7 (57.1%) infants had hearing impairments, including one with unilateral and three with bilateral hearing impairments. Two (28.6%) and 3 (42.9%) infants had epilepsy requiring antiepileptic drugs (cases 4 and 7) and cerebral palsy (cases 4, 7, and 8), respectively. Three infants (42.9%) had severe developmental delays (cases 4, 7, and 8).

### 3.2. Prenatal Imaging Findings, Postnatal Clinical Findings, and Outcomes for Fetuses with cCMV without FGR

Ten fetuses without FGR were prenatally diagnosed with cCMV based on positive CMV-DNA PCR testing of the amniotic fluid. Table 2 shows the prenatal US findings, fetal brain MRI findings, and neonatal clinical findings and outcomes. Case 4, who was referred to our hospital at 23 GW because of liver dysfunction, was positive for serum CMV IgM. Case 4 was diagnosed with mild ventriculomegaly at 28 GW and underwent a CMV-DNA PCR analysis for the amniotic fluid. The remaining 9 fetuses were referred to our hospital because of US findings associated with cCMV, including 3 cases with ventriculomegaly (cases 1, 3, and 5) and 7 cases with ascites (cases 2, 3, and 6–10), together with positive results for maternal serum CMV IgM.

All ten mothers of the fetuses were positive for CMV IgM during pregnancy. Seven mothers of the fetuses (cases 2, 4, and 6–10) were estimated to have primary CMV infections in the first or early second trimester, because they had low AI scores in the second trimester. The timing of the CMV infection was unknown in the three mothers of the fetuses (cases 1, 3, and 5).

The prevalence rates of fetal US findings associated with cCMV in the 10 cases were as follows: ventriculomegaly, 90.0% (9/10); intracranial calcification, 10.0% (1/10); cysts or pseudocysts in the brain, 0% (0/10); microcephaly, 0% (0/10); ascites, 70.0% (7/10); hepatomegaly, 80.0% (8/10); splenomegaly, 10.0% (1/10); hyperechoic bowel, 30.0% (3/10); placentomegaly 40.0% (4/10). Two fetuses had ventriculomegaly only (cases 1 and 5), whereas the remaining 8 fetuses had multiple US findings associated with cCMV. All fetuses received Ig fetal therapy except case 10, whose parents refused fetal therapy.

The fetal brain MRI findings evaluated for all cases in this group except for case 8, for which undetermined results were obtained, were as follows: ventriculomegaly, 88.9% (8/9); pseudocysts, 22.2% (2/9); cortical malformation, 11.1% (1/9); cerebellar hypoplasia, 11.1% (1/9); WM hyperintensity, 11.1% (1/9).

The median GW values at delivery and birth weight were 34 GW (range, 30–38 GW) and 2667 g (range, 1660–3216 g), respectively. The prevalence rates of clinical and imaging findings at birth in the 10 newborns with cCMV without FGR were as follows: ventriculomegaly, 90.0% (9/10); intracranial calcification, 50.0% (5/10); cysts or pseudocysts in the brain, 60.0% (6/10); microcephaly, 0% (0/10); ascites, 50.0% (5/10); hepatomegaly, 50.0% (5/10); splenomegaly, 10% (1/10); anemia, 10.0% (1/10); thrombocytopenia, 60.0% (6/10). One newborn died 1 h after birth due to having a hypoplastic lung (case 2). Eight infants received neonatal therapy, whereas case 4 did not because the infant only had mild liver dysfunction.

Four of the 8 (50.0%) live infants had hearing impairments, including two with unilateral and two with bilateral hearing impairments. The neurological outcomes of one infant (case 10) were not evaluated because this case dropped out before the 1.5-year follow-up. One of the 8 (12.5%) infants who underwent neurodevelopmental assessment after 1.5 years old had both cerebral palsy and severe developmental delay (case 9).

### 3.3. Comparison of Clinical Characteristics between the FGR and Non-FGR Groups

Table 3 shows the maternal characteristics and prenatal US findings of the 8 FGR and 10 non-FGR groups of fetuses with cCMV. The maternal age, gravidity, and parity rates did not vary between the two groups. Regarding the prenatal US findings for fetuses and placenta, the proportions of cases who had hepatomegaly (*p* = 0.054) and placentomegaly (*p* = 0.09) in the FGR group were lower than those in the non-FGR group, although not significantly. The prevalence rates of ventriculomegaly, intracranial calcification, cysts or pseudocysts in the brain, microcephaly, ascites, splenomegaly, and hyperechoic bowel did not differ between the two groups.

The clinical characteristics and outcomes of the 18 newborns with prenatally diagnosed cCMV are shown in Table 4. The birth weight in the FGR group (median 1842 g, range 1255–2192 g) was significantly lower than that in the non-FGR group (median 2667 g, range 1660–3216; *p* < 0.01). The proportion of newborns who had cysts or pseudocysts in the brain in the FGR group was lower than that in the non-FGR group but not significantly (12.5% vs. 60.0%, *p* = 0.07). The GW values at birth, cesarean delivery rates, and proportions of newborns who had physical and imaging findings of cCMV, except for cysts or pseudocysts in the brain, did not differ between the two groups. Regarding the neonatal outcomes, the incidence rates of death, bilateral hearing impairment, epilepsy, cerebral palsy, and severe developmental delay did not vary significantly between the two groups.

### 3.4. Comparison of the Diagnostic Accuracy of Prenatal US Findings between the FGR and Non-FGR Groups

The diagnostic accuracy of fetal US findings for morphological abnormalities associated with cCMV in newborns with prenatally confirmed cCMV is described in Table 5. The diagnostic sensitivity of prenatal US examinations for fetal ascites in the two groups was as high as 100.0%. In contrast, the diagnostic sensitivity rates for cysts and pseudocysts in the brain (0% in both groups), splenomegaly (0% in both groups), and intracranial calcification (20.0% in both groups) were low for both groups. The diagnostic sensitivity for hepatomegaly in the FGR group was lower than that in the non-FGR group (40.0% vs. 100.0%).

## 4. Discussion

This retrospective cohort study examined 18 fetuses who were prenatally diagnosed with cCMV via CMV-DNA PCR testing of the amniotic fluid and imaging findings associated with cCMV at our institution during the last 13 years. We found that the diagnostic sensitivity of prenatal US for identifying hepatomegaly in newborns with both cCMV and FGR was lower than that in those with cCMV without FGR (40.0% vs. 100.0%). We also found that intracranial calcification, cysts or pseudocysts in the brain, and splenomegaly in fetuses with cCMV were difficult to detect using fetal US, regardless of the presence of FGR. In general, clinicians and sonographers suspect cCMV when multiple US morphological abnormalities associated with cCMV are observed [7]. In addition, previous studies have suggested that maternal serological tests for TORCH infections, including CMV, should be considered in the presence of multiple fetal US abnormalities [10,11,12,13]. The results of the present study indicate that the detection of FGR in fetal US examinations may be accompanied by other fetal US findings of cCMV (e.g., intracranial calcification, cysts/pseudocysts in the brain, hepatomegaly, and splenomegaly). Indeed, we experienced a case that was initially diagnosed with prenatal FGR only but was found postnatally to have ventriculomegaly, intracranial calcification, hepatomegaly, and splenomegaly (case 3 in Table 1).

In addition, sonographers who were not advised about the suspicion of cCMV were more likely to overlook US findings of cCMV [14,15,16]. In the present study, all fetuses were confirmed to have cCMV prenatally, and all clinicians who performed fetal US examinations were made aware of this. Thus, morphological abnormalities caused by cCMV in many fetuses with FGR may be overlooked in a prenatal US screening.

Hepatomegaly and splenomegaly are generally diagnosed when the fetal liver and spleen lengths are beyond the 95% CI of reference values for the GW, respectively [22,23]. However, the reference values for the fetal liver and spleen lengths for the GW may not be applicable to fetuses with FGR. In fact, in the present study, the proportion of newborns diagnosed postnatally with hepatomegaly in the FGR group was higher than that in the non-FGR group (62.5% vs. 50.0%). However, the proportion of fetuses suspected of hepatomegaly on fetal US in the former was lower than that in the latter (25.0% vs. 80.0%). In particular, measuring the spleen length accurately is difficult because of the similar echogenicity levels between the fetal spleen and liver, making the margin of the spleen nearly indistinguishable. Chaoui et al. [29] proposed that the displacement by the enlarged spleen of the stomach from its left lateral position to the middle anterior position was useful for diagnosing fetal splenomegaly. However, their diagnostic US imaging criteria for fetal splenomegaly seem subjective and at some risk of observer bias. Novel diagnostic criteria for hepatomegaly and splenomegaly in FGR fetuses are necessary for the accurate prenatal risk assessment for cCMV.

On the other hand, small lesions in the fetal crania, such as calcification and cysts or pseudocysts, are more clearly observed when the ultrasonic probes are directly placed on the anterior fontanelle of the newborns than when placed on the abdominal wall of the mother. This may have lowered the diagnostic sensitivity of prenatal US for calcification and cysts or pseudocysts in the fetal brain.

In the present study, 15 of the 18 cases (except cases 7 and 8 in Table 1 and case 10 in Table 2) received Ig fetal therapy. Furthermore, ascites in three fetuses (case 4 in Table 1 and cases 2 and 10 in Table 2), hepatomegaly in 3 (cases 4, 8, and 9 in Table 2), and ventriculomegaly in 2 (case 6 in Table 1 and case 4 in Table 2) disappeared at birth. Leruez-Ville et al. [30] found that fetal therapy with oral high-dosage valacyclovir decreased the prevalence of symptomatic cCMV in the newborns who were symptomatic in utero by 82%. They also found that 43% of the fetuses who had symptomatic cCMV in utero were asymptomatic at birth in the natural course of intrauterine CMV infection. Based on the results of their trial, the Ig fetal therapy may have alleviated ascites, hepatomegaly, and ventriculomegaly in some cases in our study.

A retrospective survey of 2624 medical facilities in Japan revealed that FGR was the most predictive factor of poor outcomes in infants with cCMV [31]. It is known that CMV replicates in cytotrophoblasts and that CMV infection leads to abnormal development and function of the placenta by inhibiting cytotrophoblast differentiation and invasion [32]. This CMV-related impairment of the placental function may cause FGR, and may also strongly contribute to disease severity. It is speculated that congenitally CMV-infected fetuses with FGR have more severe placental damage, hypoxia, and malnutrition than those without FGR, meaning the neurological sequelae in the former fetuses are more severe than those in the latter fetuses. However, in the present study, the prevalence of severe long-term sequelae (e.g., bilateral hearing impairment, epilepsy, cerebral palsy, and severe developmental delay) in the fetuses with both cCMV and FGR was higher than in those without FGR, albeit non-significantly. Prospective cohort studies with larger case numbers are necessary to confirm the adverse effects of FGR in cCMV on neonatal neurological outcomes.

Fetal MRI testing provides additional information about the lesions in the fetal brain because it can detect WM alternations, abnormal gyrations, and abnormal myelination, which could not be detected by US [33,34]. Cannie et al. classified MRI findings in fetuses with cCMV into 5 grades as follows: grade 1 for normal findings; grades 2 and 3 for the presence of isolated frontal or parieto-occipital (i.e., grade 2) and temporal (i.e., grade 3) periventricular T2-weighted signal hyperintensity; grade 4 for the presence of cysts or septa in the temporal or occipital lobe; grade 5 for the presence of migration disorders, cerebellar hypoplasia, or microcephaly [35]. Previous studies demonstrated that this MRI grading was predictive of sensorineural hearing loss (NHSL) and of neurological impairment [35,36]. Cannie et al. found that 66.7% (4/6) of cases who had MRI findings in grade 5 had NHSL and neurological impairment [35]. In our study, MW hyperintensity and cortical malformation were observed in 3 (cases 4 and 8 in Table 1 and case 6 in Table 2) and 1 fetus (case 5 in Table 2), respectively. However, the diagnostic sensitivity of fetal MRI for cCMV in FGR fetuses was unclear. Conversely, cerebellar hypoplasia was detected via fetal MRI testing in all three fetuses who had cerebral palsy in the FGR group (cases 4, 7, and 8 in Table 1). Thus, fetal MRI may be useful for predicting neurological outcomes in FGR fetuses with cCMV.

On the other hand, in our study, cerebellar hypoplasia was not defined as an US finding associated with cCMV. Therefore, the perinatologists who performed US examinations in this study measured a transcerebellar diameter (TCD) only in the first US examination. In the three fetuses with FGR where cerebellar hypoplasia was detected by fetal MRI testing but not by US (cases 4, 7, and 8 in Table 1), their TCDs at the first US examination point were above the 5% CI of the reference values for GW, and serial measurements of TCD were not performed. As described previously, cerebellar hypoplasia is one of the most predictive MRI findings of neurological sequelae in fetuses with cCMV [35,36]; therefore, clinicians should examine the presence of fetal cerebellar hypoplasia during every targeted US examination for fetuses with cCMV.

It has been believed that symptoms in congenitally CMV-infected infants born to mothers with primary CMV infection during pregnancy are more severe than those with non-primary infection. Cavoretto et al. reported two cases of cCMV born to the mothers with non-primary CMV infection [37]. They suggested that immunosuppressive agents (azathioprine) may cause the reactivation of latent CMV and fetal infection. In our study, the mothers of three cases with cCMV and FGR and 7 with cCMV without FGR experienced primary maternal infection during pregnancy, because they had low CMV IgG AI rates during pregnancy. In the remaining cases, the timing of the maternal CMV infection was unknown. However, recent studies have demonstrated that the number and severity of symptoms in congenitally CMV-infected infants of mothers with non-primary CMV infection are similar or superior to those in congenitally infected infants of mothers with primary CMV infection [27,38]. 

Our retrospective cohort study suggests that the diagnostic sensitivity of fetal US examinations for hepatomegaly caused by cCMV is lower in fetuses with FGR than that in those without FGR, even when the clinicians are aware of cCMV. Imaging criteria based on the measurements of body parts (e.g., the liver, spleen, and lateral ventricular atrium) and placentas in FGR cases may be inadequate because the reference or cutoff values in the healthy controls are inapplicable to fetuses with FGR. In a previous study, the fetal liver length was measured in the 98 fetuses with small-for-gestational-age (SGA), which was defined as a fetal abdominal circumference (AC) < the 10th percentile for GW [39]. Only twenty-five percent of the fetuses with SGA had beyond the means of liver length for GW, and only 3% of them had liver lengths > the 90th percentile for GW. In another study, the fetal liver volume (FLV), which was measured by manually tracing the margin of the fetal liver on each image slice in the MRI, in 32 fetuses with cCMV was compared with 33 healthy fetuses [40]. There were no significant differences in FLV between the fetuses with cCMV and healthy fetuses, but the FLV/fetal AC ratio and FLV/fetal body volume ratio in the fetuses with cCMV were significantly higher than those in the healthy fetuses. These findings indicate that US evaluations of fetal hepatomegaly in fetuses with cCMV and FGR require adjustments based on the fetal size, such as the AC and EFBW.

The strengths of our study were the longitudinal, detailed, and systemic survey of fetuses with prenatally confirmed cCMV and the detailed assessment of long-term neurological outcomes in children with cCMV. Conversely, the present study had some limitations. This retrospective study enrolled a small number of cases, and lacked the measurements of TCD in the ultrasonographic follow-up for fetal cerebellar hypoplasia. Further prospective studies with larger datasets, including the data for TCDs, are required. Our study only included fetuses who were prenatally diagnosed with cCMV; therefore, we could not evaluate the diagnostic sensitivity for detecting fetal US findings of cCMV under conditions where clinicians and sonographers were unaware of cCMV. In addition, Ig fetal therapy and neonatal therapy with VGCV could influence the neurological outcomes of the infants. Thus, we could not assess which fetal US findings were associated with adverse outcomes of fetuses with cCMV. However, a prospective cohort study evaluating the diagnostic efficacy of prenatal US examinations for cCMV and the outcomes of affected newborns is ongoing as of the time of writing.

In conclusion, the presence of FGR may be associated with adverse neurological outcomes in fetuses with cCMV; therefore, clinicians should consider the possibility of cCMV when they encounter cases with early FGR. Importantly, clinicians and sonographers should keep in mind the possibility of overlooking the symptoms of cCMV when they perform fetal US examinations for FGR fetuses. This study may provide helpful information for clinicians and reinforce the importance of MRI as a complementary tool in assessing the fetal brain in congenital CMV infection.

## Figures and Tables

**Figure 1 diagnostics-13-00306-f001:**
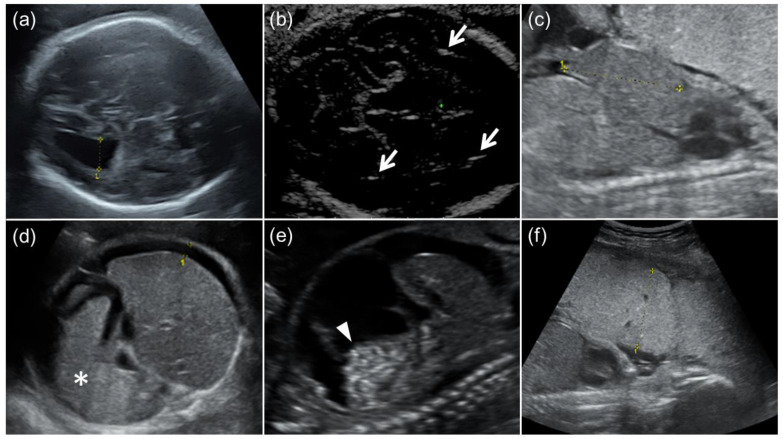
Fetal ultrasound findings of congenital cytomegalovirus (CMV) infection. (**a**) Moderate ventriculomegaly. The lateral ventricular atrial diameter at 31 GW was 1.3 cm. (**b**) Arrows indicate intracranial calcification. (**c**) Hepatomegaly. The fetal liver length at 24 GW was 4.2 cm (the reference value for 24 GW: ≤3.8 cm). (**d**) Splenomegaly and fetal ascites. An asterisk indicates enlarged fetal spleen, and the width of fetal ascites around the fetal liver was 0.8 cm. (**e**) Fetal ascites and hyperechoic bowel (arrowhead). (**f**) Placentomegaly. The placental thickness at 24 GW was 5.1 cm.

**Figure 2 diagnostics-13-00306-f002:**
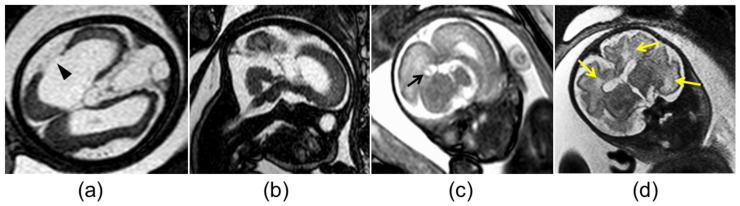
Abnormal magnetic resonance imaging findings in the brain of fetuses with congenital cytomegalovirus infection: (**a**) cortical malformation and schizencephaly (arrowhead); (**b**) vermian hypoplasia; (**c**) atrium pseudocyst (black arrow); (**d**) white matter hyperintensity (yellow arrows).

**Figure 3 diagnostics-13-00306-f003:**
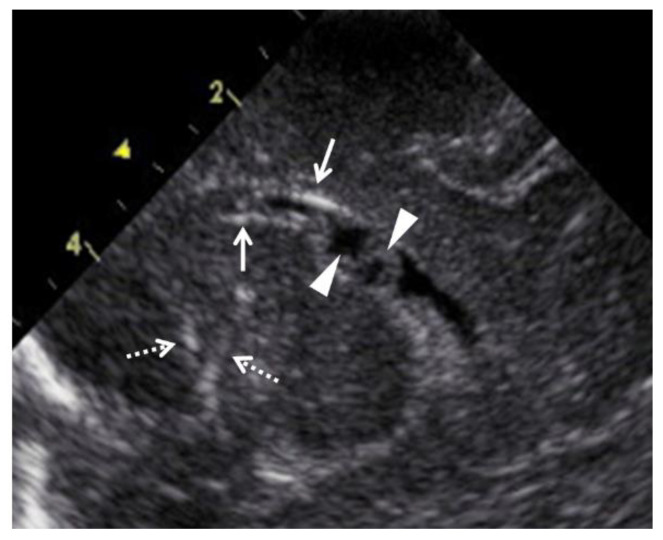
Abnormal ultrasound findings in the brain of newborns with congenital cytomegalovirus infection. Arrows indicate periventricular calcification, and arrowheads indicate pseudocysts in the cerebral ventricle. Dashed arrows indicate lenticulostriate vasculopathy.

**Table 1 diagnostics-13-00306-t001:** Prenatal and postnatal findings and outcomes of 8 fetuses with both congenital cytomegalovirus infection and fetal growth restriction.

Case	Age, Gravidity/Parity	Fetal US Findings									MRI Findings in the Fetal Brain	Clinical Findings at Birth									Outcomes					
		FGR	Ventri-culo-megaly *	Intra-cranial Calcifi-cation	Cysts/Pseudo-cysts	Micro-cephaly	Ascites	Hepato-megaly	Spleno-megaly	Others		Birthweight, Delivery Mode, (GW)	Ventri-culo-megaly	Intra-Cranial Calcifi-ccation	Cysts/Pseudo-cysts	Micro-cephaly	Ascites	Hepato-megaly	Spleno-megaly	Others	Death	Hearling Impairment	Epilepsy	Cerebral Palsy	Overall-DQ	Age at Evalua-tion
1	27, 2/1	+ (19)	−	−	−	−	+ (19)	−	−	Cardiomegaly (22), poly-hydramnios (30)	None (27)	1824 g, CS, (31)	+	+	−	−	+	−	−	Hypoplastic lung, anemia, thrombo- cytopenia	+	N/A	N/A	N/A	N/A	1 d
2	28, 4/1	+ (25)	−	−	−	−	−	−	−	Oligo-hydramnios (31)	None (31)	1396 g, CS, (32)	−	−	−	−	−	−	−	None	−	−	−	−	≥80	5 y
3	19, 3/0	+ (26)	−	−	−	−	−	−	−	None	None (28)	1378 g, CS, (31)	+	+	−	−	−	+	+	Petechia, anemia, thrombo-cytopenia, liver dysfunction	−	+ Unilateral	−	−	≥80	4 y
4	36, 1/0	+ (23)	+++ (23)	−	−	+ (30)	+ (21)	−	−	None	Ventriculomegaly, pseudocyst, schizencephaly, cerebellar hypoplasia, WM hyperintensity (36)	2184 g, CS, (36)	+	+	−	−	−	+	+	Schizencephaly, thrombocytopenia, liver dysfunction, chorioretinitis	−	+ Bilateral	+	+	<70	3 y
5	23, 2/1	+ (19)	++ (25)	−	−	−	−	+ (25)	−	Hyperechoic bowel (21), cardiomegaly (22)	Ventriculomegaly (30)	2192 g, CS, (36)	+	−	−	−	−	+	−	Petechia, thrombo-cytopenia, cholestasis	−	−	−	−	70–79	5 y
6	32, 8/2	+ (29)	++ (33)	−	−	−	−	−	−	None	Ventriculomegaly, pseudocyst (34)	2030 g, CS, (36)	−	+	+	−	−	−	−	Petechia, thrombo-cytopenia, liver dysfunction	−	−	−	−	≥80	4 y
7	22, 2/0	+ (30)	+ (35)	+ (30)	−	+ (30)	−	+ (35)	−	None	Cerebellar hypoplasia (31)	1860 g, CS, (36)	+	+	−	+	−	+	+	Petechia, thrombo-cytopenia, liver dysfunction, retinal vascular malformation	−	+ Bilateral	+	+	<70	3 y
8	28, 1/0	+ (23)	+ (34)	−	−	−	−	−	−	None	Ventriculomegaly, cerebellar hypoplasia, WM hyperintensity (31)	1255 g, CS, (35)	+	−	−	+	−	+	−	WM lesion, cerebral atrophy, cholestasis, thrombocytopenia, liver dysfunction	−	+ Bilateral	−	+	<70	3 y

Number in ( ) indicates gestational weeks at an identification of each US finding or MRI findings. * Mild (+), moderate (++), or severe (+++) ventriculomegaly in US examinations. Abbreviations: US, ultrasound; MRI, magnetic resonance imaging; GW, gestational weeks; FGR, fetal growth restriction; DQ, developmental quotient; CS, cesarean section; N/A, not applicable; WM, white matter.

**Table 2 diagnostics-13-00306-t002:** Prenatal and postnatal findings and outcomes of 10 fetuses with congenital cytomegalovirus infection but without fetal growth restriction.

Case	Age, Gravidity/Parity	Fetal US Findings									MRI Findings in the Fetal Brain	Clinical Findings at Birth									Outcomes					
		FGR	Ventri-culo-megaly *	Intra-cranial calcifi-cation	Cysts/Pseudo-cysts	Micro-cephaly	Ascites	Hepato-megaly	Spleno-megaly	Others		Birthweight, Delivery Mode, (GW)	Ventri-culo-megaly	Intra-cranial Calcifi-cation	Cysts/Pseudo-cysts	Micro-cephaly	Ascites	Hepato-megaly	Spleno-megaly	Others	Death	Hearling Impairment	Epilepsy	Cerebral Palsy	Overall-DQ	Age at Evalua-tion
1	35, 2/1	−	+++ (30)	−	−	−	−	−	−	None	Ventriculomegaly, pseudocyst (33)	2956 g, VD, (38)	+	−	+	−	−	−	−	None	−	+ Unilateral	−	−	≥ 80	5 y
2	30, 3/1	−	+ (24)	−	−	−	+ (22)	+ (24)	+ (24)	Cardiomegaly (24), peri-cardial effusion (24), palacento-megaly (24)	None (30)	2236 g, CS, (31)	+	+	−	−	−	+	−	Hypoplastic lung, petechia, anemia, thrombo-cytopenia, liver dysfunction	+	N/A	N/A	N/A	N/A	0 d
3	21, 1/0	−	++ (28)	−	−	−	+ (28)	+ (28)	−	Hyperechoic bowel (28)	Ventriculomegaly, pseudocyst (31)	2688 g, CS, (33)	+	+	+	−	+	+	+	Petechia, thrombo-cytopenia, cholestasis	−	+ Bilateral	−	−	≥80	3 y
4	29, 1/0	−	+ (28)	−	−	−	−	+ (35)	−	None	Ventriculomegaly (32)	2996 g, CS, (37)	−	−	−	−	−	−	−	Liver dysfunction	−	−	−	−	≥80	3 y
5	37, 2/1	−	+++ (30)	−	−	−	−	−	−	None	Ventriculomegaly, cortical malformation (32)	2646 g, VD, (38)	+	−	+	−	−	−	−	None	−	−	−	−	≥80	3 y
6	29, 2/0	−	++ (24)	+ (24)	−	−	+ (20)	+ (24)	−	Placento- megaly (24)	Ventriculomegaly, MW hyperintensities, cerebellar hypoplasia (34)	2848 g, CS, (34)	+	+	+	−	+	+	−	Petechia, thrombo-cytopenia	−	−	−	−	≥80	3 y
7	29, 1/0	−	+ (24)	−	−	−	+ (21)	+ (22)	−	None	Ventriculomegaly (31)	2312 g, CS, (33)	+	−	+	−	+	+	−	Petechia, thrombo-cytopenia	−	+ Unilateral	−	−	≥80	3 y
8	19, 1/0	−	+ (24)	−	−	−	+ (23)	+ (24)	−	Hyperechoic bowel (24), placento-megaly (24)	U/D (24)	1660 g, CS, (32)	+	+	−	−	+	−	−	Petechia, neutropenia, thrombo-cytopenia	−	−	−	−	70–79	3 y
9	32, 2/1	−	−	−	−	−	+ (20)	+ (27)	−	Hyperechoic bowel (20), placento-megaly (22)	Ventriculomegaly (30)	1860 g, CS, (30)	+	+	−	−	+	−	−	Petechia, thrombo-cytopenia, liver dysfunction	−	−	−	+	<70	1.5 y
10	30, 3/2	−	+ (23)	−	−	−	+ (20)	+ (34)	−	None	Ventriculomegaly (30)	3216 g, VD, (38)	+	−	+	−	−	+	−	None	−	+ Bilateral	−	N/D	N/D	Dropped out before 1.5y

Number in ( ) indicates gestational weeks at an identification of each US finding or MRI findings. * Mild (+), moderate (++), or severe (+++) ventriculomegaly in US examinations. Abbreviations: US, ultrasound; FGR, fetal growth restriction; MRI, magnetic resonance imaging; GW, gestational weeks; DQ, developmental quotient; VD, vaginal delivery; CS, cesarean section; U/D, undetermined.

**Table 3 diagnostics-13-00306-t003:** Maternal characteristics and prenatal imaging findings of 18 fetuses with congenital cytomegalovirus infection.

		All n = 18	cCMV with FGR n = 8	cCMV without FGR n = 10	*p*-Value
Maternal characteristics					
	Age	29 (19–37)	28 (19–36)	30 (19–37)	0.3
	Gravidity	2 (1–8)	2 (1–8)	2 (1–3)	0.3
	Parity	1 (0–2)	1 (0–2)	1 (0–2)	1.0
US findings of the fetuse or placenta					
	Ventriculomegaly				
	≥mild	14 (77.8%)	5 (62.5%)	9 (90.0%)	0.3
	≥moderate	7 (38.9%)	3 (37.5%)	4 (40.0%)	1.0
	≥severe	3 (16.7%)	1 (12.5%)	2 (20.0%)	1.0
	Intracranial calcification	2 (11.1%)	1 (12.5%)	1 (10.0%)	1.0
	Cysts/pseudocysts in the brain	0 (0%)	0 (0%)	0 (0%)	1.0
	Microcephaly	2 (11.1%)	2 (25.0%)	0 (0%)	0.2
	Ascites	9 (50.0%)	2 (25.0%)	7 (70.0%)	0.2
	Hepatomegaly	10 (55.6%)	2 (25.0%)	8 (80.0%)	0.054
	Splenomegaly	1 (5.6%)	0 (0%)	1 (10.0%)	1.0
	Hyperechoic bowel	4 (22.2%)	1 (12.5%)	3 (30.0%)	0.6
	Placentomegaly	4 (22.2%)	0 (0%)	4 (40.0%)	0.09

Data are expressed as the median (range) or number (percentage). Abbreviations: cCMV, congenital cytomegalovirus infection; FGR, fetal growth restriction; US, ultrasound.

**Table 4 diagnostics-13-00306-t004:** Clinical characteristics and outcomes of 18 newborns with congenital cytomegalovirus infection.

	All n = 18	cCMV with FGR n = 8	cCMV without FGR n = 10	*p*-Value
Neonatal characteristics				
GW at birth	35 (30–38)	36 (31–36)	34 (30–38)	0.7
Birth weight (g)	2188 (1255–3216)	1842 (1255–2192)	2667 (1660–3216)	0.006
Cesarean delivery	15 (83.3%)	8 (100%)	7 (70%)	0.2
Physical and imaging findings of newborns				
Ventriculomegaly	15 (83.3%)	6 (75.0%)	9 (90.0%)	0.6
Intracranial calcification	10 (55.6%)	5 (62.5%)	5 (50.0%)	0.7
Cysts/pseudocysts in the brain	7 (38.9%)	1 (12.5%)	6 (60.0%)	0.07
Microcephaly	2 (11.1%)	2 (25.0%)	0 (0%)	0.2
Ascites	6 (33.3%)	1 (12.5%)	5 (50.0%)	0.2
Hepatomegaly	10 (55.6%)	5 (62.5%)	5 (50.0%)	0.7
Splenomegaly	4 (22.2%)	3 (37.5%)	1 (10.0%)	0.3
Anemia	3 (16.7%)	2 (25.0%)	1 (10.0%)	0.6
Thrombocytopenia	13 (72.2%)	7 (87.5%)	6 (60.0%)	0.3
Adverse outcomes in affected newborns				
Death	2 (11.1%)	1 (12.5%)	1 (10.0%)	1.0
Bilateral hearing impairment	5 (31.3% ^a^)	3 (42.9% ^b^)	2 (22.2% ^c^)	0.6
Epilepsy	2 (12.5% ^a^)	2 (28.6% ^b^)	0 (0% ^c^)	0.2
Cerebral palsy	4 (26.7% ^d^)	3 (42.9% ^b^)	1 (12.5% ^e^)	0.3
Overall DQ < 70	4 (26.7% ^d^)	3 (42.9% ^b^)	1 (12.5% ^e^)	0.3

Data are expressed as the median (range) or number (percentage). ^a^ Two newborns who died were excluced; ^b^,^c^ One newborns who died was excluded; ^d^ Two newborns who died and 1 who was dropped out were excludded; ^e^ One dead and 1 dropout cases were excluded. Abbreviations: cCMV, congenital cytomegalovirus infection; FGR, fetal growth restriction; GW, gestational weeks; DQ, developmental quotient.

**Table 5 diagnostics-13-00306-t005:** Diagnostic accuracy of prenatal ultrasound examinations for detecting morphological abnormalities associated with congenital cytomegalovirus infection.

Findings Associated with cCMV	Subjects	Sensitivity (%)	Specificity (%)	Positive Predictive Value (%)	Negative Predictive Value (%)	Accuracy (%)
Ventriculomegaly	cCMV with FGR (n = 8)	66.7	50.0	80.0	33.3	62.5
	cCMV without FGR (n = 10)	88.9	0	88.9	0	80.0
	All (n = 18)	80.0	33.3	85.7	25.0	72.2
Intracranial calcification	cCMV with FGR (n = 8)	20.0	100.0	100.0	42.9	50.0
	cCMV without FGR (n = 10)	20.0	100.0	100.0	55.6	60.0
	All (n = 18)	20.0	100.0	100.0	50.0	55.6
Cysts/pseudocysts	cCMV with FGR (n = 8)	0	100.0	N/A	87.5	87.5
	cCMV without FGR (n = 10)	0	100.0	N/A	50.0	50.0
	All (n = 18)	0	100.0	N/A	66.7	66.7
Microcephaly	cCMV with FGR (n = 8)	50.0	83.3	50.0	83.3	75.0
	cCMV without FGR (n = 10)	N/A	100.0	N/A	100.0	100.0
	All (n = 18)	50.0	93.8	50.0	93.8	88.9
Ascites	cCMV with FGR (n = 8)	100.0	85.7	50.0	100.0	87.5
	cCMV without FGR (n = 10)	100.0	60.0	71.4	100.0	80.0
	All (n = 18)	100.0	75.0	66.7	100.0	83.3
Hepatomegaly	cCMV with FGR (n = 8)	40.0	100.0	100.0	50.0	62.5
	cCMV without FGR (n = 10)	100.0	40.0	62.5	100.0	70.0
	All (n = 18)	70.0	62.5	70.0	62.5	66.7
Splenomegaly	cCMV with FGR (n = 8)	0	100.0	N/A	62.5	62.5
	cCMV without FGR (n = 10)	0	88.9	0	88.9	80.0
	All (n = 18)	0	92.9	0	76.5	72.2

Abbreviations: cCMV, congenital cytomegalovirus infection; FGR, fetal growth restriction; N/A, not available.

## Data Availability

Not applicable.

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
