# Peer review of "Fetal Ultrasound and Magnetic Resonance Imaging Abnormalities in Congenital Cytomegalovirus Infection Associated with and without Fetal Growth Restriction"

_diagnostics, 2023, doi:10.3390/diagnostics13020306_

Round 1
Reviewer 1 Report
To compare the cCMV infection in fetuses with fetal growth restriction. This manuscript is only 18 acses to survey the cCMV inefction with well follow up during prenatal and postnatal for a couple years. It is icluding US findings the fetal growth restriction (FGR), ventriculomegaly, intracranial calcification, cysts or pseudocysts in the brain, microcephaly, ascites, hepatosplenomegaly, hyperechoic bowel, and placentomegaly. Beside, the MR images examination also be used by the magnetic resonance imaging (MRI) to assessment the fetuses. Both of diagnostic methods well describe in this manuscipt.
Author Response
Point-by-point response to reviewers’ comments
Reviewer 1
Overall comment: To compare the cCMV infection in fetuses with fetal growth restriction. This manuscript is only 18 cases to survey the cCMV infection with well follow up during prenatal and postnatal for a couple years. It is including US findings the fetal growth restriction (FGR), ventriculomegaly, intracranial calcification, cysts or pseudocysts in the brain, microcephaly, ascites, hepatosplenomegaly, hyperechoic bowel, and placentomegaly. Besides, the MR images examination also be used by the magnetic resonance imaging (MRI) to assessment the fetuses. Both of diagnostic methods well describe in this manuscript.
Response: We thank the reviewer for quite positively evaluating our findings and providing valuable comments.
Reviewer 2 Report
I want to thank the Multidisciplinary Digital Publishing Institute and Ms. Aubrey Zhang, assistant editor, for the opportunity to review this article.
I would also like to thank the authors of this article for their effort and for their pursuit to improve the diagnostic accuracy of prenatal imaging of fetuses infected with cytomegalovirus.
I have read the article by Tanimura et al. entitled “Diagnostic accuracy of fetal ultrasound for abnormalities associated with congenital cytomegalovirus infection in fetuses with fetal growth restriction.
The authors present the results of a retrospective cohort of 18 fetuses with CMV infection diagnosed prenatally and their follow-up at a single center, a tertiary referral university hospital in Japan, for over 12 years.
This manuscript shows an enormous amount of work and effort.
Overall, the manuscript is interesting, read well, and adequately written. Results are satisfactory, and conclusions are derived from the results. Nevertheless, there are essential concerns regarding this paper.
General:
In this study, the findings in fetal growth-restricted fetuses with CMV infection are exciting and much-needed data needing reporting and investigating.
Please make sure to correct the minor spelling errors throughout the manuscript.
Title:
Since their sample size is inadequate to refer to this paper as one of diagnostic accuracy, I strongly recommend to please change the title as follows:
Fetal ultrasound and magnetic resonance imaging abnormalities in congenital cytomegalovirus infection associated with and without fetal growth restriction.
Introduction:
Line 67: change PCR of the amniotic fluid to PCR in amniotic fluid.
Objective:
Please change it to:
Describe the associated fetal ultrasound/MRI abnormalities in CMV-infected fetuses with and without FGR. In addition, we compared the diagnostic performance of targeted US examinations according to the FGR status to determine if there are differences in US findings associated with cCMV in fetuses with growth restriction.
Methods:
Please look at the article for the definition of fetal growth restriction used in this paper and refer to it.
I recommend using the definition of estimated fetal weight below the 3rd centile for the gestational age as well and include the following references:
- Consensus definition of fetal growth restriction: a Delphi procedure. Ultrasound Obstet Gynecol 2016 Sep;48(3):333-9. doi: 10.1002/uog.15884.
- Validation of Delphi procedure consensus criteria for defining fetal growth restriction. Ultrasound Obstet Gynecol 2020; 56: 61–66. doi: 10.1002/uog.20854
Line 95: please include the following article as a reference:
Placental enlargement in women with primary maternal cytomegalovirus infection is associated with fetal and neonatal disease. Clinical Infectious Diseases 2006; 43:994-1000.
Line 100: Please change as follows: (e) fetal ascites and hyperechogenic bowel (arrowhead)
Line 101: In figure 1, please put the measures of the ventricle, right hepatic lobe, ascites, and placentomegaly.
Line 115: In figure 2, please correct as follows: (a) Cortical malformation and schizencephaly (arrowhead).; (b) vermian hypoplasia.; (c) atrium pseudocyst (black arrow), please change this image for an axial or sagittal view.; (d) white matter hyperintensity (yellow arrows). If you can find a better image, please change it.
Line 147: In figure 3, please signal the lenticulostriate vasculopathy. Please change arrow heads and pseudo cysts to “arrowheads and pseudocysts.”
Results:
Please describe the time and type of infection of the study population. Were these cases first-trimester primary infections, second-trimester seroconversions, re-infections, or unknown?
Line 182: Please use hearing impairment instead of hearing difficulties. Please change it in the entire manuscript.
Line 188: Please erase the word “but.”
Table 4. Please use the word impairment instead of difficulties.
It could be possible to have a composite outcome called “adverse perinatal outcome” and make the comparison. There is a high probability that you will find differences.
Discussion:
Line 334: the fact that there were three fetuses in the FGR in which cerebellar hypoplasia was diagnosed by the MRI and not with the US-targeted examination should be discussed. These cerebellar hypoplasias were missed due to the ultrasound examiner’s inexperience in the fetal brain or differences in the reference values used in each imaging modality.
Line 336: Please elaborate more regarding this issue. Fetal MRI is a complementary tool for predicting neurological outcomes in CMV-infected fetuses.
Please use the following papers to elaborate more:
- Revisiting short- and long-term outcome after fetal first-trimester primary cytomegalovirus infection in relation to prenatal imaging findings. Ultrasound Obstet Gynecol 2020; 56: 572–578. Doi: 10.1002/uog.21946
- Congenital cytomegalovirus infection: contribution and best timing of prenatal MR imaging. Eur Radiol 2016 Oct;26(10):3760-9. Doi: 10.1007/s00330-015-4187-0.
Line 338: please discuss the possibility of the limitation for accurately estimating an enlarged fetal liver implied in a 2D measurement.
Discuss the possible hypotheses related to this finding (hepatomegaly is lower in fetuses with FGR than in those without FGR)
Recommend the following articles:
- Roberts AB, Mitchell JM, McCowan LM, Barker S. Ultrasonographic measurement of liver length in the small-for-gestational-age fetus. Am J Obstet Gynecol. (1999) 180:634–8. Doi: 10.1016/S0002-9378(99)70266-8
- Fetal liver volume assessment using magnetic resonance imaging in fetuses with cytomegalovirus infection. Doi: 10.3389/fmed.2022.889976
Please discuss that the neurological outcome of the FGR infants was worse than the non-FGR infants. The FGR per se probably increases the risk of the poor neurodevelopmental outcome observed. Considering that 88% (7/8) of FGR fetuses were diagnosed before 29 weeks of gestation, they likely were infected before 14 weeks. Thus, this has been the cause of their poor neurodevelopmental outcome.
Please elaborate more on your strengths and limitations.
Please put the conclusion after strengths and limitations and add that according to your results, fetal growth restriction may be a US finding that increases the risk of the poor neurodevelopmental outcome.

Author Response
Point-by-point response to reviewers’ comments
Reviewer 2
Overall comment: The authors present the results of a retrospective cohort of 18 fetuses with CMV infection diagnosed prenatally and their follow-up at a single center, a tertiary referral university hospital in Japan, for over 12 years. This manuscript shows an enormous amount of work and effort. Overall, the manuscript is interesting, read well, and adequately written. Results are satisfactory, and conclusions are derived from the results. Nevertheless, there are essential concerns regarding this paper.
Response: We thank the reviewer for quite positively evaluating our findings and providing valuable comments.
Comment on General: In this study, the findings in fetal growth-restricted fetuses with CMV infection are exciting and much-needed data needing reporting and investigating.
Please make sure to correct the minor spelling errors throughout the manuscript.
Response: We thank the reviewer for giving us a kind advice. This manuscript has been edited for English language, grammar, punctuation, and spelling by Enago©. I’ll attach a certification of editing from Enago©.
Comment on the Title: Since their sample size is inadequate to refer to this paper as one of diagnostic accuracy, I strongly recommend to please change the title as follows:
Fetal ultrasound and magnetic resonance imaging abnormalities in congenital cytomegalovirus infection associated with and without fetal growth restriction.
Response: We thank you very much for your valuable comment. We changed the title according to the reviewer’s suggestion.
Comment on the Introduction: Line 67: change PCR of the amniotic fluid to PCR in amniotic fluid.
Response: We have corrected the part (Line 67) according to the reviewer’s suggestion.
Comment on the Objective: Please change it to: Describe the associated fetal ultrasound/MRI abnormalities in CMV-infected fetuses with and without FGR. In addition, we compared the diagnostic performance of targeted US examinations according to the FGR status to determine if there are differences in US findings associated with cCMV in fetuses with growth restriction.
Response: According to the reviewer’s suggestion, in Lines 69–73, the objective was changed to “we described the associated fetal ultrasound/MRI abnormalities in CMV-infected fetuses with and without FGR. In addition, we compared the diagnostic performance of targeted US examinations according to the FGR status to determine if there are differences in US findings associated with cCMV in fetuses with growth restriction.”
Comment on the Methods: Please look at the article for the definition of fetal growth restriction used in this paper and refer to it. I recommend using the definition of estimated fetal weight below the 3rd centile for the gestational age as well and include the following references:
-Consensus definition of fetal growth restriction: a Delphi procedure. Ultrasound Obstet Gynecol 2016 Sep;48(3):333-9. doi: 10.1002/uog.15884.
-Validation of Delphi procedure consensus criteria for defining fetal growth restriction. Ultrasound Obstet Gynecol 2020; 56: 61–66. doi: 10.1002/uog.20854
Response: According to the reviewer’s suggestion, the sentence in the Lines 89-90 was changed to “FGR was defined as an estimated fetal body weight < the 3rd percentile for gestational weeks (GW) [18, 19].
And, we added the following two references.
- Gordijn, S.J.;Beune, I.M.;Thilaganathan, B.;Papageorghiou, A.;Baschat, A.A.;Baker, P.N.;Silver, R.M.;Wynia, K.; Ganzevoort, W. Consensus definition of fetal growth restriction: a Delphi procedure. Ultrasound Obstet Gynecol. 2016,48,333-339.
- Molina, L.C.G.;Odibo, L.;Zientara, S.;Obican, S.G.;Rodriguez, A.;Stout, M.; Odibo, A.O. Validation of Delphi procedure consensus criteria for defining fetal growth restriction. Ultrasound Obstet Gynecol. 2020,56,61-66.
Fortunately, the change in the definition of FGR did neither influence the grouping of the cases nor the results in this study.
Comment on the Methods: Line 95: please include the following article as a reference:
Placental enlargement in women with primary maternal cytomegalovirus infection is associated with fetal and neonatal disease. Clinical Infectious Diseases 2006; 43:994-1000.
Response: According to the reviewer’s suggestion, we added the following reference (Line 96).
- La Torre, R.;Nigro, G.;Mazzocco, M.;Best, A.M.; Adler, S.P. Placental enlargement in women with primary maternal cytomegalovirus infection is associated with fetal and neonatal disease. Clin Infect Dis. 2006,43,994-1000.
Comment on the Methods: Line 100: Please change as follows: (e) fetal ascites and hyperechogenic bowel (arrowhead)
Response: We have corrected the part (Lines 103–104) according to the reviewer’s suggestion.
Comment on the Methods: Line 101: In figure 1, please put the measures of the ventricle, right hepatic lobe, ascites, and placentomegaly.
Response: According to the reviewer’s suggestion, we changed the legend of Figure 1 (Lines 99–104) to “Figure 1. Fetal ultrasound findings of congenital cytomegalovirus (CMV) infection. (a) moderate ventriculomegaly. The lateral ventricular atrial diameter at 31GW was 1.3cm.; (b) arrows indicate intracranial calcification.; (c) hepatomegaly. The fetal liver length at 24GW was 4.2cm (the reference value for 24GW: ≤ 3.8cm).; (d) splenomegaly and fetal ascites. An asterisk indicates enlarged fetal spleen, and the width of fetal ascites around the fetal liver was 0.8cm.; (e) fetal ascites and hyperechoic bowel (arrowhead).; (f) placentomegaly. The placental thickness at 24GW was 5.1cm.
Comment on the Methods: Line 115: In figure 2, please correct as follows: (a) Cortical malformation and schizencephaly (arrowhead).; (b) vermian hypoplasia.; (c) atrium pseudocyst (black arrow), please change this image for an axial or sagittal view.; (d) white matter hyperintensity (yellow arrows). If you can find a better image, please change it.
Response: According to the reviewer’s suggestion, we changed the legend of Figure 2 (Lines 199–121) to “Figure 2. Abnormal magnetic resonance imaging findings in the brain of fetuses with congenital cytomegalovirus infection. (a) Cortical malformation and schizencephaly (arrowhead).; (b) vermian hypoplasia.; (c) atrium pseudocyst (black arrow).; (d) white matter hyperintensity (yellow arrows).”
Unfortunately, we cannot find any images showing atrium pseudocyst for an axial or sagittal view. In addition, we cannot find a better MRI image showing white matter hyperintensity than the present one. I am very sorry that we cannot change the images of Figure 2 (c) and (d).
Comment on the Methods: Line 147: In figure 3, please sign the lenticulostriate vasculopathy. Please change arrow heads and pseudo cysts to “arrowheads and pseudocysts.”
Response: According to the reviewer’s suggestion, we added dashed arrows which indicate lenticulostriate vasculopathy into Figure 3. In addition, we changed the sentences in the legend of Figure 3 (Lines 151–154) to “Arrows indicate periventricular calcification, and arrowheads do pseudocysts in the cerebral ventricle. Dashed arrows indicate lenticulostriate vasculopathy.”
Comment on the Results: Please describe the time and type of infection of the study population. Were these cases first-trimester primary infections, second-trimester seroconversions, re-infections, or unknown?
Response: We thank the reviewer for giving important suggestions. According to reviewer’s suggestion, in Lines 167–174, “All eight mothers of the fetuses were positive for CMV IgM during pregnancy. Two mothers of the fetuses (Cases 1 and 3) were estimated to have primary CMV infection in or before the second trimester, because they had low CMV-IgG avidity index (AI) (Aisenkai Nichinan Hospital Miyazaki, Japan) (cutoff value: <35% [27]) in the second trimester. One mother of Case 2, who had low CMV IgG AI in the third trimester, could have primary CMV infection in or before the third trimester. In the five mothers, the timing of CMV infection was unknown, because they had high CMV IgG AI in the third trimester (Cases 4,5, and 6) or did not receive avidity measurements (Cases 7 and 8).” was added.
And, in Lines 211–215, “All ten mothers of the fetuses were positive for CMV IgM during pregnancy. Seven mothers of the fetuses (Cases 2, 4, and 6–10) were estimated to have primary CMV infection in or before the second trimester, because they had low AI in the second trimester. In the three mothers of the fetuses (Cases 1, 3, and 5), the timing of CMV infection was unknown, because they had high AI in the third trimester.” was added.
And, we added the following reference.
- Tanimura, K.;Tairaku, S.;Morioka, I.;Ozaki, K.;Nagamata, S.;Morizane, M.;Deguchi, M.;Ebina, Y.;Minematsu, T.; Yamada, H. Universal Screening With Use of Immunoglobulin G Avidity for Congenital Cytomegalovirus Infection. Clin Infect Dis. 2017,65,1652-1658.
Comment: Line 182: Please use hearing impairment instead of hearing difficulties. Please change it in the entire manuscript.
Response: According to the reviewer’s suggestion, we corrected “hearing difficulties” in the text and the Tables 1, 2, and 4 to “hearing impairment”.
Comment on the Results: Line 188: Please erase the word “but.”
Response: According to the reviewer’s suggestion, we erased the “but” (Line 200).
Comment on the Results: Table 4. Please use the word impairment instead of difficulties.
Response: According to the reviewer’s suggestion, we corrected “hearing difficulties” in the Tables 1, 2, and 4 to “hearing impairment”.
Comment on the Results: It could be possible to have a composite outcome called “adverse perinatal outcome” and make the comparison. There is a high probability that you will find differences.
Response: We thank the reviewer for giving us a kind advice. We compared the incidence of all possible combinations of adverse outcomes in affected fetuses/newborns between the FGR group and the non-FGR group. However, unfortunately, we could not find any significant differences between the two groups. We think that the number of cases in both groups was too small to find significant differences.
Comment on the Discussion: Line 334: the fact that there were three fetuses in the FGR in which cerebellar hypoplasia was diagnosed by the MRI and not with the US-targeted examination should be discussed. These cerebellar hypoplasias were missed due to the ultrasound examiner’s inexperience in the fetal brain or differences in the reference values used in each imaging modality.
Response: We thank the reviewer for giving us an important suggestion. We didn’t select fetal cerebellar hypoplasia as an US finding associated with cCMV in this study. Therefore, perinatologists who performed fetal US examinations could have detected fetal cerebellar hypoplasia, if they had examined the presence of it at every US examination.
In Lines 371–380, “On the other hand, in our study, cerebellar hypoplasia was not defined as an US finding associated with cCMV. Therefore, the perinatologists who performed US examinations in this study measured a transcerebellar diameter (TCD) only at the first US examination. In the three fetuses with FGR where cerebellar hypoplasia was detected by fetal MRI but not by US (Cases 4, 7, and 8 in Table 1), their TCDs at the first US examination were beyond the 5% CI of the reference values for GW, and serial measurements of TCD were not performed. As describe previously, cerebellar hypoplasia is one of the most predictive MRI findings of neurological sequelae in fetuses with cCMV [34, 35]; therefore, clinicians should examine the presence of fetal cerebellar hypoplasia at every targeted US examination for fetuses with cCMV.” was added.
Comment on the Discussion: Line 336: Please elaborate more regarding this issue. Fetal MRI is a complementary tool for predicting neurological outcomes in CMV-infected fetuses.
Please use the following papers to elaborate more:
- Revisiting short- and long-term outcome after fetal first-trimester primary cytomegalovirus infection in relation to prenatal imaging findings.Ultrasound Obstet Gynecol 2020; 56: 572–578. Doi: 10.1002/uog.21946
- Congenital cytomegalovirus infection: contribution and best timing of prenatal MR imaging.Eur Radiol 2016 Oct;26(10):3760-9. Doi: 10.1007/s00330-015-4187-0.
Response: We thank the reviewer for giving us very important advices. According to reviewer’s suggestion, in Lines 356–364, “Cannie et al. classified MRI findings in fetuses with cCMV into 5 grades as follows: grade 1 for normal findings, grades 2 and 3 for the presence of isolated frontal or parieto-occipital (i.e., grade 2) and temporal (i.e., grade 3) periventricular T2-weighted signal hyperintensity, grade 4 for the presence of cysts and/or septa in the temporal and/or occipital lobe, and grade 5 for the presence of migration disorders, cerebellar hypoplasia, microcephaly [35]. Previous studies demonstrated that this MRI grading was predictive of sensorineural hearing loss (NHSL) and of neurological impairment [35, 36]. Especially, Cannie et al. found that 66.7% (4/6) of cases who had MRI findings in the grade 5 had NHSL and neurological impairment [35].” was added.
And, we added the following two references.
- Cannie, M.M.;Devlieger, R.;Leyder, M.;Claus, F.;Leus, A.;De Catte, L.;Cossey, V.;Foulon, I.;Van der Valk, E.;Foulon, W.; et al. Congenital cytomegalovirus infection: contribution and best timing of prenatal MR imaging. Eur Radiol. 2016,26,3760-3769.
- Lipitz, S.;Elkan Miller, T.;Yinon, Y.;Weissbach, T.;De-Castro, H.;Hoffman, C.;Katorza, E.; Weisz, B. Revisiting short- and long-term outcome after fetal first-trimester primary cytomegalovirus infection in relation to prenatal imaging findings. Ultrasound Obstet Gynecol. 2020,56,572-578.
Comment on the Discussion: Line 338: please discuss the possibility of the limitation for accurately estimating an enlarged fetal liver implied in a 2D measurement.
Discuss the possible hypotheses related to this finding (hepatomegaly is lower in fetuses with FGR than in those without FGR)
Recommend the following articles:
- Roberts AB, Mitchell JM, McCowan LM, Barker S. Ultrasonographic measurement of liver length in the small-for-gestational-age fetus. Am J Obstet Gynecol. (1999) 180:634–8. Doi: 10.1016/S0002-9378(99)70266-8
- Fetal liver volume assessment using magnetic resonance imaging in fetuses with cytomegalovirus infection.Doi: 10.3389/fmed.2022.889976
Response: We thank the reviewer for giving us very valuable advices. According to reviewer’s suggestion, in Lines 398–409, “In a previous study, fetal liver length was measured in the 98 fetuses with small-for-gestational-age (SGA), which was defined as a fetal abdominal circumference (AC) < 10th percentile for GW [39]. Only twenty-five percent of the fetuses with SGA had beyond the means of liver length for GW, and only 3% of them had liver length of > 90th percentile for GW. In another study, fetal liver volume (FLV), which was measured by manually tracing the margin of fetal liver on each image slice in MRI, in the 32 fetuses with cCMV was compared with that in the 33 healthy fetuses [40]. There were no significant differences in FLV between the fetuses with cCMV and healthy fetuses, but the FLV/fetal AC ratio and FLV/fetal body volume ratio in the fetuses with cCMV were significantly higher than those in the healthy fetuses. These findings indicate that US evaluation of fetal hepatomegaly in fetuses with cCMV and FGR requires an adjustment by fetal size, for example, AC and EFBW.” was added.
And, we added the following two references.
- Roberts, A.B.;Mitchell, J.M.;McCowan, L.M.; Barker, S. Ultrasonographic measurement of liver length in the small-for-gestational-age fetus. Am J Obstet Gynecol. 1999,180,634-638.
- Hawkins-Villarreal, A.;Moreno-Espinosa, A.L.;Martinez-Portilla, R.J.;Castillo, K.;Hahner, N.;Nakaki, A.;Trigo, L.;Picone, O.;Siauve, N.;Figueras, F.; et al. Fetal Liver Volume Assessment Using Magnetic Resonance Imaging in Fetuses With Cytomegalovirus Infection(dagger). Front Med (Lausanne). 2022,9,889976.
Comment on the Discussion: Please discuss that the neurological outcome of the FGR infants was worse than the non-FGR infants. The FGR per se probably increases the risk of the poor neurodevelopmental outcome observed. Considering that 88% (7/8) of FGR fetuses were diagnosed before 29 weeks of gestation, they likely were infected before 14 weeks. Thus, this has been the cause of their poor neurodevelopmental outcome.
Response: We thank the reviewer for giving us very valuable advice. As we described in the revised Results section, the time of maternal CMV infection in the 3 cases who had severe neurological sequelae in the FGR group (Cases 4, 7, and 8 in Table 1) were unknown, because they had high CMV-IgG AI in the third trimester (Case 4) or did not receive AI measurements during pregnancy (Case 7 and 8). Therefore, we cannot say that earlier timing of maternal primary infection is associated with severity of neurological sequelae in the FGR group. Therefore, in Lines 341–348,” It is known that CMV replicates in cytotrophoblasts, and CMV infection leads to abnormal development and function of the placenta by inhibiting cytotrophoblasts differentiation and invasion [32]. Such CMV-related impairment of placental function may cause FGR, and may also strongly contribute to disease severity. It is speculated that congenitally CMV-infected fetuses with FGR have more severe placental damage, hypoxia, and malnutrition than those without FGR, and thus the neurological sequelae in the former fetuses are more severe than those in the latter fetuses.” was added.
And, we added the following reference.
- Tabata, T.;Petitt, M.;Zydek, M.;Fang-Hoover, J.;Larocque, N.;Tsuge, M.;Gormley, M.;Kauvar, L.M.; Pereira, L. Human cytomegalovirus infection interferes with the maintenance and differentiation of trophoblast progenitor cells of the human placenta. J Virol. 2015,89,5134-5147.
Comment on the Discussion: Please elaborate more on your strengths and limitations.
Response: According to the reviewer’s suggestion, in Lines 410–412,” The strengths of our study were the longitudinal, detailed, and systemic survey of fetuses with prenatally confirmed cCMV, and the detailed assessment of long-term neurological outcomes in children with cCMV.” and in Lines 417–419,” The two radiologists retrospectively evaluated the MRI images, a prospective cohort study to evaluate the accuracy of fetal MRI in predicting neurological outcomes in fetuses with cCMV is required.” were added.
Comment on the Discussion: Please put the conclusion after strengths and limitations and add that according to your results, fetal growth restriction may be a US finding that increases the risk of the poor neurodevelopmental outcome.
Response: According to reviewer’s suggestion, in Lines 425–430,” In conclusion, the presence of FGR may be associated with adverse neurological outcomes in fetuses with cCMV, clinicians therefore should consider the possibility of cCMV when they encounter cases with FGR. Importantly, clinicians and sonographers should keep in mind the possibility of overlooking the symptoms of cCMV when they perform fetal US examinations for FGR fetuses. This result may provide useful information to clinicians. “was added as a conclusion of our study.

Round 2
Reviewer 2 Report
I want to thank the authors of this article for their effort and for improving their manuscript.
I have read the revised article by Tanimura et al. entitled “Fetal ultrasound and magnetic resonance imaging abnormalities in congenital cytomegalovirus infection associated with and without fetal growth restriction.”
This manuscript shows an enormous amount of work and effort.
Overall, the manuscript is interesting, read well, and adequately written. Results are satisfactory, and conclusions are derived from the results. Nevertheless, there are minor concerns regarding this reviewed version.
General:
Please make sure to correct the minor spelling errors throughout the manuscript.
Abstract:
Include in line 30 before the conclusion the following:
The prevalence of severe long-term sequelae (e.g., bilateral hearing impairment, epilepsy, cerebral palsy, and severe developmental delay) in the CMV-infected fetuses with FGR was higher, albeit non-significantly.
Results:
Line 172: Please change the sentence as follows:
One mother of Case 2, who had low CMV IgG AI in the third trimester, could have primary CMV infection in the second trimester.
Line 173: Please change the sentence as follows:
In the five mothers, the timing of CMV infection was unknown.
Line 211-213: Please change the sentence as follows:
Seven mothers of the fetuses (Cases 2, 4, and 6–10) were estimated to have primary CMV infection in the first or early second trimester.
Line 213-215:
The timing of CMV infection was unknown in the three mothers of the fetuses (Cases 1, 3, and 5).
Discussion:
Line 341: Please change “for poor outcomes” to “of poor outcomes.”
Line 351: Please correct as follows: without FGR, albeit non-significantly.
Line 376: Please change “beyond the 5% CI” to “above the 5% CI”
Line 413: include the retrospective nature of the analysis.
Please include in your limitations the lack of the measurement of the TCD in the ultrasonographic follow-up.
Line 417-419:
This line is difficult to understand. Erase it. Include this limitation in line 413.
“Retrospective nature of the analysis.”
Conclusion:
Line 427: Please include early FGR.
Line 429: Please change it to This study may provide helpful information to clinicians and reinforce the importance of MRI as a complementary tool in assessing the fetal brain in cCMV infection.

Author Response
Point-by-point response to reviewers’ comments
Reviewer 2
Overall comment: This manuscript shows an enormous amount of work and effort.
Overall, the manuscript is interesting, read well, and adequately written. Results are satisfactory, and conclusions are derived from the results. Nevertheless, there are minor concerns regarding this reviewed version.
Response: We thank the reviewer for quite positively evaluating our findings and providing valuable comments.
Comment on the General: Please make sure to correct the minor spelling errors throughout the manuscript.
Response: We thank the reviewer for giving kind advice. We have checked spelling throughout the manuscript.
Comment on the Abstract: Include in line 30 before the conclusion the following: The prevalence of severe long-term sequelae (e.g., bilateral hearing impairment, epilepsy, cerebral palsy, and severe developmental delay) in the CMV-infected fetuses with FGR was higher, albeit non-significantly.
Response: We thank the reviewer for giving us a kind advice. In Lines 30–32, “The prevalence of severe long-term sequelae (e.g., bilateral hearing impairment, epilepsy, cerebral palsy, and severe developmental delay) in the CMV-infected fetuses with FGR was higher, albeit non-significantly.” was added.
Comment on the Results: Line 172: Please change the sentence as follows: One mother of Case 2, who had low CMV IgG AI in the third trimester, could have primary CMV infection in the second trimester.
Response: According to the reviewer’s suggestion, we changed the part in Lines 174–175 to “One mother of Case 2, who had low CMV IgG AI in the third trimester, could have primary CMV infection in the second trimester.”
Comment on the Results: Line 173: Please change the sentence as follows: In the five mothers, the timing of CMV infection was unknown.
Response: According to the reviewer’s suggestion, we changed the part in Lines 175–176 to “In the five mothers (Cases 4–8), the timing of CMV infection was unknown.”
Comment on the Results: Line 211-213: Please change the sentence as follows: Seven mothers of the fetuses (Cases 2, 4, and 6–10) were estimated to have primary CMV infection in the first or early second trimester.
Response: According to the reviewer’s suggestion, we changed the part in Lines 213–215 to “Seven mothers of the fetuses (Cases 2, 4, and 6–10) were estimated to have primary CMV infection in the first or early second trimester, “
Comment on the Results: Line 213-215: The timing of CMV infection was unknown in the three mothers of the fetuses (Cases 1, 3, and 5).
Response: According to the reviewer’s suggestion, we changed the part in Lines 216–217 to “The timing of CMV infection was unknown in the three mothers of the fetuses (Cases 1, 3, and 5).”
Comments on the Discussion:
Line 341: Please change “for poor outcomes” to “of poor outcomes.”
Line 351: Please correct as follows: without FGR, albeit non-significantly.
Line 376: Please change “beyond the 5% CI” to “above the 5% CI”
Response: We have corrected these parts (Lines 343, 353, and 378) according to the reviewer’s suggestion.
Comment on the Discussion: Line 413: include the retrospective nature of the analysis.
Please include in your limitations the lack of the measurement of the TCD in the ultrasonographic follow-up.
Response: According to the reviewer’s suggestion, we changed the part in Lines 414–416 to “This retrospective study enrolled a small number of cases, and lacked the measurements of TCD in the ultrasonographic follow-up for fetal cerebellar hypoplasia. Further prospective studies with larger datasets, including the data of TCDs, are required.”
Comment on the Discussion: Line 417-419: This line is difficult to understand. Erase it. Include this limitation in line 413. “Retrospective nature of the analysis.”
Response: According to the reviewer’s suggestion, we erased the sentences and emphasized that this study was a retrospective one.
Comment on the Conclusion: Line 427: Please include early FGR.
Response: According to the reviewer’s suggestion, we changed the part in Line 427 to” early FGR”.
Comment on the Conclusion: Line 429: Please change it to This study may provide helpful information to clinicians and reinforce the importance of MRI as a complementary tool in assessing the fetal brain in cCMV infection.
Response: According to the reviewer’s suggestion, we changed the part in Lines 429–431 to”This study may provide helpful information to clinicians and reinforce the importance of MRI as a complementary tool in assessing the fetal brain in congenital CMV infection.”
